# Near-field acoustic imaging with a caged bubble

Dorian Bouchet ⓘ, Olivier Stephan ⓘ, Benjamin Dollet ⓘ, Philippe Marmottant ⓘ & Emmanuel Bossy ⓘ ✉

Bubbles are ubiquitous in many research applications ranging from ultrasound imaging and drug delivery to the understanding of volcanic eruptions and water circulation in vascular plants. From an acoustic perspective, bubbles are resonant scatterers with remarkable properties, including a large scattering cross-section and strongly sub-wavelength dimensions. While it is known that the resonance properties of bubbles depend on their local environment, it remains challenging to probe this interaction at the single-bubble level due to the difficulty of manipulating a single resonating bubble in a liquid. Here, we confine a cubic bubble inside a cage using 3D printing technology, and we use this bubble as a local probe to perform scanning near-field acoustic microscopy—an acoustic analog of scanning near-field optical microscopy. By probing the acoustic interaction between a single resonating bubble and its local environment, we demonstrate near-field imaging of complex structures with a resolution that is two orders of magnitudes smaller than the wavelength of the acoustic field. As a potential application, our approach paves the way for the development of low-cost acoustic microscopes based on caged bubbles.

Scanning near-field optical microscopy (SNOM) is an invaluable tool for the study of light-matter interaction in the optical regime[1]. Following the original proposition of Synge in 1928[2] and the development of aperture-type microscopes[3], SNOM has been demonstrated with a variety of sub-wavelength probes such as metallic tips[4], gold particles[5] and fluorescent molecules[6]. This allows not only to reconstruct images of samples with a sub-wavelength resolution[7], but also to investigate electromagnetic interactions at the nanoscale, for instance to study single-molecule fluorescence in the vicinity of photonic antennas[8,9] and to map the local density of optical states close to nanostructured materials[10–12].

For many applications in nondestructive testing and biological imaging[13,14], it is necessary to be specifically sensitive to the elastic properties of materials. In this context, it is thus relevant to use acoustic waves instead of light as a sensing mechanism. In the acoustic regime, probing near-field interactions entails using acoustic resonators instead of optical emitters. Acoustic interactions can be probed using different types of resonators such as a tuning fork[15], a cantilever[16], a Chinese gong[17], or a nanoparticle[18]. However, further

advances in scanning near-field acoustic microscopy (SNAM) are hindered by the difficulty of finding local probes that can be easily manipulated, that strongly interact with the acoustic field, and that are sub-wavelength along all dimensions. Gas bubbles are excellent candidates for this application, thanks to their large scattering cross-section and their strongly sub-wavelength dimensions[19,20]. As such, ensembles of bubbles freely flowing inside blood vessels have been used to reconstruct super-resolved images by ultrasound localization microscopy[21]—an acoustic analog of photo-activated localization microscopy[22,23]. However, in order to use a single resonating bubble as a local probe for SNAM, one needs to scan the position of the bubble in three dimensions. First steps in this direction have been achieved by attaching an oil droplet to the cantilever of an atomic force microscope[24,25], an idea that has then be applied to the characterization of interaction forces between an air bubble and its surrounding environment[26–29]. Another manipulation strategy consists in using optical tweezers to trap single bubbles, a technique that has been used to experimentally observe changes in microbubble dynamics close to interfaces[30,31].

Univ. Grenoble Alpes, CNRS, LIPhy, 38000 Grenoble, France. ✉ e-mail: emmanuel.bossy@univ-grenoble-alpes.fr

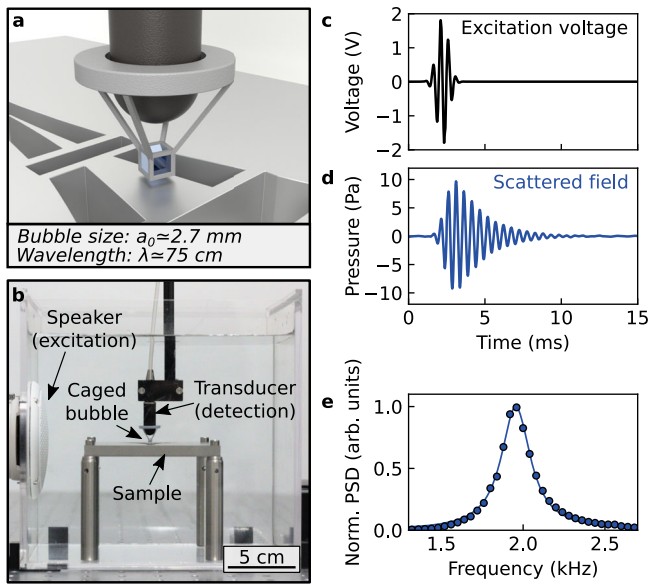

**Fig. 1 | Principle of a scanning near-field acoustic microscope based on a caged bubble. a** Artistic representation of a cubic bubble confined inside a 3D-printed cage placed at the tip of a hydrophone. **b** Photograph of the experiment. A caged bubble, whose position is controlled with a 3D motorized stage, is scanned in the near field of a structured sample. The resonance of the bubble is excited by an acoustic pulse generated using a speaker, and the scattered field is recorded by a transducer holding the 3D-printed cage. **c** Excitation voltage supplied to the speaker. **d** Field scattered by the bubble in the absence of sample. **e** Normalized power spectral density (PSD) of the field measured in the absence of sample (blue dots), along with a Lorentzian fit to the data (blue line).

Here, we harness the possibility to manipulate a resonating acoustic bubble inside a cage using 3D printing technology[32,33] and experimentally demonstrate that this bubble can be used as a local probe for SNAM (Fig. 1). By measuring variations in the resonance properties of the bubble induced by near-field acoustic interactions, we reconstruct images of structured samples with a resolution that is not limited by diffraction but by the size of the bubble, which is two order of magnitudes smaller than the wavelength of the acoustic field. The method offers different contrast mechanisms, as the various resonance parameters of the bubble (such as its resonance amplitude or its resonance frequency) are influenced by the acoustic impedance of the surrounding environment in a specific way. By demonstrating a SNAM approach based on a single resonating bubble, we thus introduce a tool to determine acoustic properties of structured materials at the sub-wavelength scale.

## Results

### Principle of the experiment

Following a mechanical excitation by an acoustic wave, a gas bubble in a liquid behaves as a resonant scatterer, as its volume oscillates about an equilibrium value. For a spherical bubble, the resonance frequency is given by the Minnaert formula $f_0 = c_g \sqrt{3\rho_g/\rho_l}/(\pi d_0)$ where $c_g$ is the speed of sound in the gas, $\rho_g$ and $\rho_l$ are the densities of the gas and the liquid, and $d_0$ is the diameter of the bubble at equilibrium[19]. At resonance, the ratio between the wavelength $\lambda$ of acoustic waves in the liquid and the diameter of the bubble is thus

$$\frac{\lambda}{d_0} = \frac{\pi c_l}{c_g} \sqrt{\frac{\rho_l}{3\rho_g}}, \qquad (1)$$

where $c_l$ is the speed of sound in the liquid. For an air bubble in water at 20 °C, Eq. (1) yields $\lambda/d_0 = 226$. As such, an air bubble in water is

inherently a strongly sub-wavelength resonant scatterer, and thus constitutes an ideal local probe for the acoustic field.

To experimentally demonstrate how a single bubble can be used for SNAM, we built a millimetric cubic cage using a 3D printing technique based on digital light processing. A commercial resin is polymerized using a 3D printer, and a water-repellent treatment is applied to ensure an efficient hydrophobicity of the cage (see section "Methods" for a detailed description of the fabrication of the cages). When immersed into a water tank, this cage confines and stabilizes an air bubble, as illustrated in Fig. 1a. Indeed, for cage lengths up to a few millimeters, the surface tension induced by the hydrophobicity of the cage overcomes the hydrostatic pressure induced by gravity, thereby preventing water from entering inside the cage. The position of such a bubble is then easily controlled in three dimensions by moving the cage using a motorized stage. The cubic geometry of the bubbles has been chosen for its simplicity of fabrication, but other bubble shapes could also be considered[34,35]. The resonance frequency of a cubic bubble of side length $a_0$ is essentially driven by the volume $V$ of the gas in the bubble, as previously investigated using bubbles of polyhedral shapes[34]. Using a diameter $d_0 = 2a_0[3/(4\pi)]^{1/3}$ in Eq. (1), we obtain $\lambda/a_0 = 280$, evidencing that cubic bubbles in water are also strongly sub-wavelength resonators, in the same way as spherical ones.

To probe this resonance experimentally, we excite the bubble externally with a broadband pulse generated by an underwater loudspeaker, and we measure the acoustic signal using a transducer located in the vicinity of the bubble (Fig. 1b, c see also section "Methods"). In all experiments, we first measure the pressure field in the presence of the bubble $\phi_w(\mathbf{r}, t)$ (i.e., with air inside the cage), and we then measure the field in the absence of the bubble $\phi_{w/o}(\mathbf{r}, t)$ (i.e., with water inside the cage), which is achieved by replacing air by water by use of a pipette. The field scattered by the bubble is then defined as $\phi_s(\mathbf{r}, t) = \phi_w(\mathbf{r}, t) - \phi_{w/o}(\mathbf{r}, t)$. From these time-resolved measurements (Fig. 1d), we then calculate the normalized power spectral density $|A_s(\mathbf{r}, \omega)|^2 = |\hat{\phi}_s(\mathbf{r}, \omega)/\hat{\phi}_{w/o}(\mathbf{r}, \omega)|^2$, which is the frequency spectrum of the scattered field deconvolved by the excitation signal. Such a spectrum is well described by a Lorentzian function (Fig. 1e), with a quality factor around 10. All experiments are performed with an external cage size of 3 mm ($V \simeq 20$ mm³, $a_0 \simeq 2.7$ mm), for which a resonance frequency of 1.9 kHz is predicted by the Minnaert formula. In practice, the observed resonance frequency typically lies between 1.9 kHz and 2.0 kHz, indicating that small deviations can occur in the volume of air trapped within the cage. Once a bubble is confined inside a cage, it remains stable for several hours, and the small decrease of air volume that occurs over long measurement times can be easily corrected for using a linear correction (see Supplementary Materials, Section S1).

### Bubble dynamics close to an interface

The resonance of a bubble is known to convey information about the acoustic impedance of its surrounding environment[20]. To demonstrate the possibility of measuring these acoustic interactions with a caged bubble in water, we study the canonical situation of a single resonating bubble whose center is located at a controlled distance $z$ from an interface. We first experimentally investigate the case of a bubble close to a water-steel interface (Fig. 2a), which approximates a Neumann boundary condition (BC) for the pressure (zero normal velocity). In this case, a negative frequency shift is observed close to the interface (Fig. 2b, c). For comparison purposes, we then investigate the case of a water-air interface (Fig. 2e), which approximates a Dirichlet BC (stress-free interface). The amplitude of the measured field close to the interface is strongly reduced (Fig. 2f), but the spectral analysis does reveal a positive frequency shift close to the interface (Fig. 2g).

Different approaches can be considered to theoretically determine variations in the resonance properties of the bubble due to the presence of the interface. One such approach is to solve a modified Rayleigh-Plesset equation describing the oscillation of a bubble near

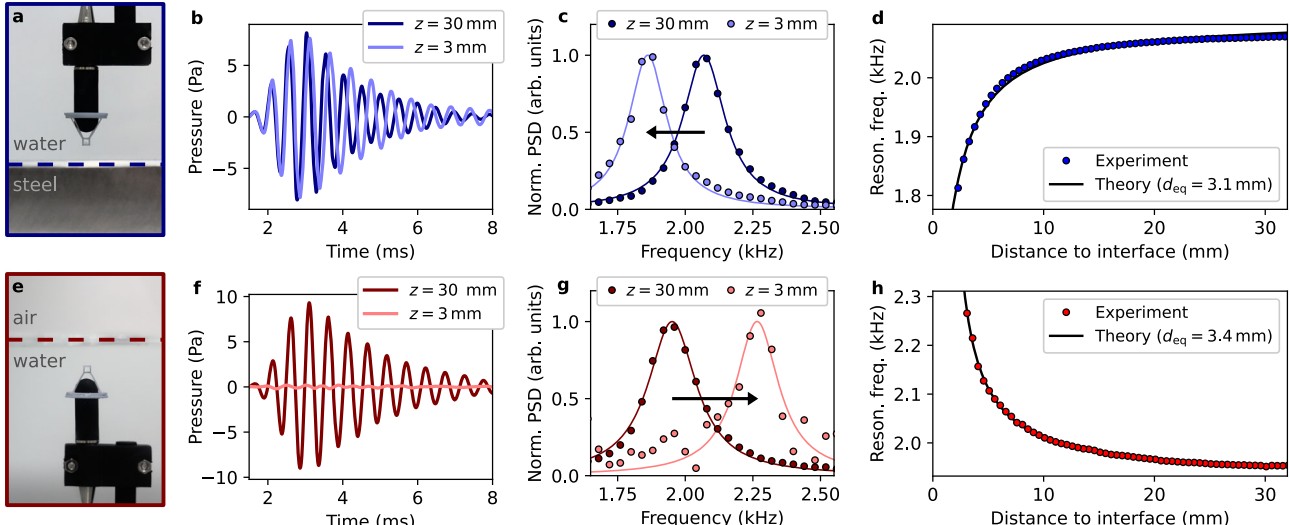

**Fig. 2 | Measurements of the bubble dynamics close to interfaces. a** Photograph of the experiment, in which a caged bubble is scanned in the vertical direction above a water-steel interface (blue dashed line). **b** Field scattered by the bubble in the vicinity of the interface ($z = 3$ mm, light blue) and far from the interface ($z = 30$ mm, dark blue). The resonance frequency is visibly reduced close to the interface. **c**, Normalized power spectral density of the scattered field when the bubble is in the vicinity of the interface ($z = 3$ mm, light blue) and far from the interface ($z = 30$ mm, dark blue). **d** Experimental measurements of the resonance

frequency of the bubble as a function of the distance to the interface (blue points), along with theoretical predictions given by Eq. (2). **e–h** Analogous to (**a–d**) for a caged bubble scanned below a water-air interface. In the vicinity of the interface, the measured field is very weak (**f**), but the frequency shift experienced by the bubble can still be observed (**g**). As theoretically predicted by Eq. (2), the behavior of the resonance frequency is different in the case of a water-air interface (the resonance frequency increases) and in the case of a water-steel interface (the resonance frequency decreases).

an elastic wall[36]. To model our experiments, in which bubbles are placed in the near field of rigid and stress-free interfaces, we will rather use the exact expression derived by Morioka based on the potential flow of an incompressible liquid around two spherical bubbles[37]:

$$f_\pm / f_0 = \sqrt{\sum_{n=0}^{+\infty} \frac{(\mp 1)^n \sinh(\beta)}{\sinh[(n+1)\beta]}}, \quad (2)$$

where $\cosh(\beta) = 2z/d_0$, and where $f_+$ and $f_-$ denote the resonance frequency of two in-phase and out-of-phase bubbles, respectively. Using the method of images, $f_+$ and $f_-$ are found to be also the resonance frequency of a single bubble close to rigid (Neuman BC) and free (Dirichlet BC) interfaces, respectively[19]. Note that, for $z \gg d_0/2$, this expression matches the approximate solution obtained by Strasberg[38], which is $f_\pm / f_0 = 1/\sqrt{1 \pm d_0/(4z)}$.

While Eq. (2) describes the case of a spherical bubble close to an interface separating two semi-infinite media, our experiments involve a cubic bubble inside a tank of finite dimensions. Therefore, in order to compare our experimental results to those predicted by the theory, we need to study how the resonance frequency of the bubble is influenced by the cubic geometry of the bubble and by the presence of the tank. For this purpose, we conducted 3D numerical simulations based on a finite-difference time-domain (FDTD) solver of the elastodynamic equations[39] (see section "Methods"). We considered the case of a tank filled with water (same dimensions as in the experiments) containing a cubic air bubble close to rigid and stress-free interfaces. As expected, we observed that the bubble radiates much more in the vicinity of a rigid interface (Fig. 3a) as compared to a stress-free interface (Fig. 3b). We then calculated the evolution of its resonance frequency as a function of the bubble-interface distance. However, in order to compare these results to the theory expressed by Eq. (2), we needed to determine an effective diameter for a cubic bubble of edge size $a_0$.

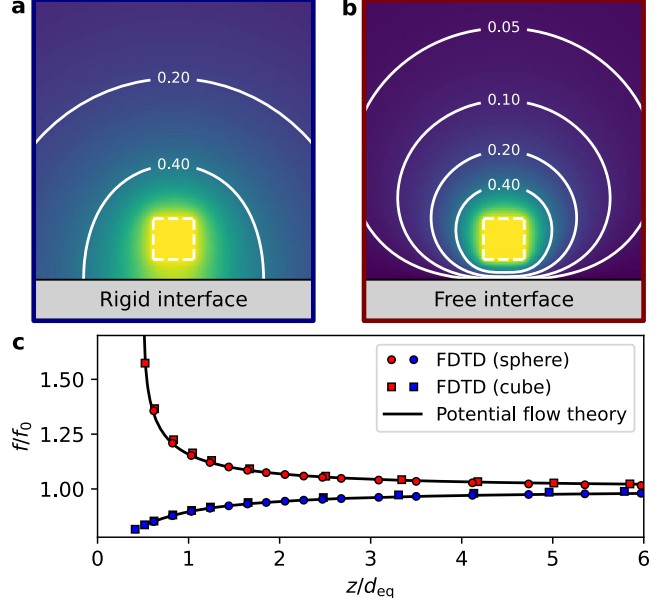

**Fig. 3 | Modeling of the bubble dynamics close to interfaces.** Normalized instantaneous pressure fields calculated by FDTD around a cubic bubble close to a rigid interface (**a**) and a stress-free interface (**b**). On these figures, the edges of the bubble are represented by dotted white lines (edge size of 2.5 mm). **c** Relation between the normalized resonance frequency $f/f_0$ and the normalized bubble-interface distance $z/d_{eq}$ for rigid (in blue) and free (in red) interfaces. Theoretical predictions obtained using Eq. (2) are represented by black lines. Results of FDTD simulations performed with a spherical bubble in water surrounding by perfectly-matched layers are represented by blue and red circles (with $d_{eq}$ being taken as the true diameter of the bubble). Results of FDTD simulations performed with a cubic bubble in water enclosed in a finite tank are represented by blue and red squares (with $d_{eq}$ obtained by fitting FDTD results to the predictions of Eq. (2)).

While taking $d_{eq} = 2a_0[3/(4\pi)]^{1/3}$ based on the volume of air enclosed in the bubble would be a good approximation (see Supplementary Materials, Section S2.1), a better agreement was obtained by considering the effective diameter as a free parameter. Using this procedure, FDTD results are accurately described by theoretical predictions obtained with Eq. (2) (Fig. 3c). In the experiments, both the effective bubble diameter and the minimal bubble-interface distance need to be treated as free parameters (see Supplementary Materials, Section 6). Using this procedure, we observed again an excellent agreement between theoretical predictions and experimental results (Fig. 2d, h). This shows that we can neglect the influence of the (non-resonant) tank, evidencing that the bubble resonance frequency is mostly sensitive to the extreme near-field of the bubble ($z \ll \lambda$). Furthermore, this also demonstrates that the resonance frequency of a cubic bubble close to an interface behaves very similarly to that of a spherical one. This can be understood from the similar pressure fields emitted by spherical and cubic bubbles (see Supplementary Materials, Section S2.2), which are essentially those that would be generated by two in-phase monopoles (rigid interface) or two out-of-phase monopoles (stress-free interface).

## Super-resolved near-field images

Beyond 1D line scans in the $z$-direction, we also demonstrate the possibility to reconstruct super-resolved 2D images by scanning the bubble in the $xy$-plane above structured samples. This procedure is illustrated in the supplementary movies, in which one can hear the sound emitted by the bubble while watching images being built. The resonance of the bubble is characterized by several parameters, including the central frequency, the linewidth, the integrated signal

energy, and the shape of the power spectrum. We can thus apply different imaging modalities, which yield different contrasts depending on the acoustic properties of the samples. Here, we perform intensity imaging, which consists in imaging the spatial dependence of the power spectral density at a given frequency (Fig. 4a). In addition, we also implement central frequency imaging, which consists in imaging the spatial dependence of the measured central frequency that is estimated from a Lorentzian fit to the power spectral density (Fig. 4b). We apply these approaches to a stainless-steel sample on which an artistic representation of the Eiffel tower is engraved (Fig. 4c). By scanning the bubble in the close vicinity of the sample (the distance between the center of the bubble and the surface of the sample is set to $z = 2.5$ mm) and by probing point-by-point the resonance of the bubble, we obtain super-resolved images of the sample (Fig. 4d, e), on which details as small as $\lambda/250$ (distance of 3 mm) can be distinguished. As expected, the measured resonance frequency decreases in the vicinity of stainless steel (Fig. 4e). Furthermore, the signal also decreases in the vicinity of stainless steel on the intensity image reconstructed at $f = 1.96$ kHz (Fig. 4d; intensity images for other frequencies are available in Supplementary Materials, Section S3.1). This is in fact a direct consequence of the frequency shift experienced by the bubble, as 1.96 kHz corresponds here to the resonance frequency of the bubble above the sample areas filled with water.

As evidenced in Fig. 2, the resonance of the bubble varies differently depending on the acoustic properties of the surrounding medium. Therefore, using a bubble as a local probe for SNAM enables one not only to recover spatial features of complex samples with a strongly sub-wavelength resolution, but also to access information about the acoustic properties of the samples. To illustrate this advantage, we

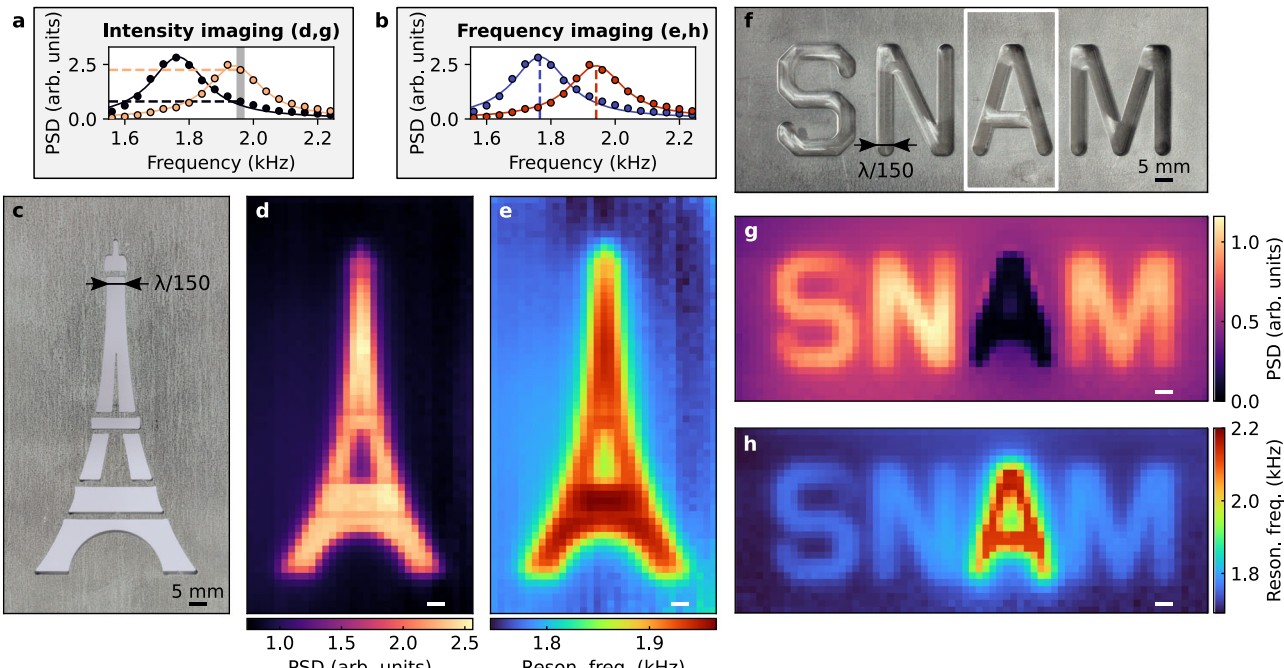

**Fig. 4 | Demonstration of super-resolution near-field imaging with a single resonating bubble. a** Illustration of the intensity imaging mode: a frequency is chosen (here $f = 1.96$ kHz) and an image is reconstructed based on the power spectral density measured at this frequency. **b** Illustration of the central frequency imaging mode: an image is reconstructed based on measured central frequencies, that are estimated from a Lorentzian fit to the power spectral densities. **c** Photograph of a stainless-steel sample on which an artistic representation of the Eiffel tower is engraved. The thickness of the plate is 8 mm, and engraved patterns go through the whole plate thickness. Super-resolved images obtained by scanning a caged bubble 2 mm above the sample shown in (**c**), in the intensity imaging mode

at $f = 1.96$ kHz (**d**) and in the central frequency mode (**e**). Instead of being limited by diffraction ($\lambda/2 \simeq 375$ mm), the resolution of the approach is given by the size of the bubble (3 mm), two orders of magnitude below the resolution limit. **f**–**h** Analogous to (**c**–**e**) for a stainless-steel sample engraved with the SNAM (scanning near-field acoustic microscopy) acronym. The thickness of the plate is 8 mm, and patterns are engraved over a thickness of 7 mm. In the experiment, the letter "A" is covered with adhesive tape (white rectangle in **f**), in order to trap an air layer in the letter. The contrast of the resulting super-resolved images (**g**, **h**) varies depending on the acoustic properties of the sample (air for the letter "A", water for other letters).

study a sample with spatially heterogeneous properties (Fig. 4f): the SNAM acronym is engraved on a stainless-steel plate, with the letter "A" covered with adhesive tape before immersing the sample in water. In this way, this letter is filled with air, while other letters are filled with water. This difference in acoustic properties can clearly be observed on the reconstructed super-resolved images (Fig. 4g, h). As expected, the measured resonance frequency decreases in the vicinity of stainless steel and increases in the vicinity of air (Fig. 4h). In contrast, the signal decreases in the vicinity of air on the intensity image measured at $f = 1.96$ kHz (Fig. 4g; intensity images for other frequencies are available in Supplementary Materials, Section S3.2). This is explained not only by the shift in the resonance frequency, but also by the decrease in the integrated signal energy measured from the bubble.

### Transverse resolution

It is not straightforward to provide a quantitative measure of the resolution of the technique. The resolution of an image is often defined from the 2D point response function of the system[40], but this function is not unique in our case as it depends on the third dimension of the system (the sample thickness) as well as on the imaging mode (intensity imaging or frequency imaging). For this reason, we adopt here an alternative approach inspired by Fourier ring correlation (FRC), a method which is commonly used in different research fields including cryo-electron microscopy[41,42] and single-molecule localization microscopy[43,44]. This approach provides a systematic way to identify the maximal spatial frequency $\xi$ for which the signal is significantly larger than the noise. The approach is implemented directly from raw experimental data, which are in our case the pressure fields measured in the presence of the bubble. Here, our resolution test sample is composed of three lines engraved in a stainless-steel plate, and we

measure the resonance frequency of the bubble for different distances $z$ (Fig. 5a). Spatial variations become less visible for larger values of $z$, which qualitatively evidences that the resolution rapidly degrades with the bubble-sample distance. This is confirmed quantitatively by our spectral analysis inspired by FRC (see Supplementary Materials, Section S4), which reveals a decrease of the maximal spatial frequency $\xi$ with $z$ (Fig. 5b) and therefore a degradation of the associated spatial resolution $\mathcal{R} = 1/\xi$ (Fig. 5c).

The relation between the distance $z$ and the resolution $\mathcal{R}$ can be understood based on the angular spectrum representation of the pressure field[45]:

$$\phi_w(\mathbf{r}, t) = \frac{1}{4\pi^2} \iint \phi_w(k_x, k_y, z = 0, t) \exp(i\mathbf{k} \cdot \mathbf{r}) \mathrm{d}k_x \mathrm{d}k_y, \quad (3)$$

where $\phi_w(k_x, k_y, z = 0, t)$ is the spatial Fourier transform of $\phi_w(x, y, z = 0, t)$ along the transverse dimensions ($x$ and $y$), and where $\| \mathbf{k} \|^2 = k_x^2 + k_y^2 + k_z^2 = (2\pi/\lambda)^2$. Deeply sub-wavelength structured samples are described by transverse wavenumbers $k_\parallel = \sqrt{k_x^2 + k_y^2}$ that satisfy $k_\parallel \gg 2\pi/\lambda$, which results in $k_z \simeq ik_\parallel$ and therefore $\exp(ik_z z) \simeq \exp(-k_\parallel z)$. Consequently, in the deeply near-field regime, high transverse spatial frequencies are exponentially attenuated through propagation in the $z$ direction, with an attenuation factor that scales with the transverse wavenumber. This is confirmed by our spectral analysis of experimental data, which shows that the maximal spatial frequency $\xi$ scales with $1/z$ (Fig. 5b) and that the measured resolution $\mathcal{R}$ scales with $z$ (Fig. 5c). In the close vicinity of the sample (for $z = 2.5$ mm), the measured resolution is approximately 3 mm, which is on the order of the bubble size.

### Discussion

While these experiments were performed here with millimetric bubbles for the sake of experimental simplicity, we emphasize that our proposed approach is inherently scalable: indeed, for a gas bubble in liquid, the size of the bubble is always much smaller than the wavelength of the scattered waves. This can be seen from Eq. (1), since typically $\rho_l \gg \rho_g$ and $c_l \gg c_g$. On the one hand, scaling up the size of the bubble cage in the centimeter range[34] would enable locally-resolved rheological measurements of interfaces in the few hundred Hz frequency range, otherwise impossible with diffraction-limited transducers which would be several meters in size. On the other hand, our approach could be implemented on the micrometer scale by engineering cages via 3D microfabrication[46], which would allow to reach a micrometer resolution using conventional MHz electronics and transducers. Current commercial acoustic microscopes, widely used for non-destructive testing in the industry and biomedical applications on the micrometer scale[14,47], are based on expensive electronics and transducers working in the GHz range. Our approach would enable acoustic microscopy with micrometer resolution at the cost of an MHz ultrasound imaging system, that are typically one or two orders of magnitude cheaper than GHz systems. Towards this goal, a challenge will be to stabilize micrometer-scale bubble and counteract the loss of air dissolving into the water, which could be done either with a coating or using a gaz injection system with a micro-capillary. The sensitivity of the transducer will also be an important aspect, as the scattering cross-section will be much smaller than those of millimetric bubbles. However, this is not expected to be a critical issue; indeed, with sensitive transducers and amplifiers, it has already been demonstrated that it is possible to detect single micrometric bubbles, not only in controlled experiments[31] but also in vivo through the skull[21]. Note that a standard focused single-element MHz transducer could be used to both excite and detect sound from the bubble probe, which would result in a more compact setup as compared to our proof-of-concept setup with a hydrophone and a speaker.

The method inherently offers various contrast mechanisms, as the different resonance parameters of the bubble are influenced by the

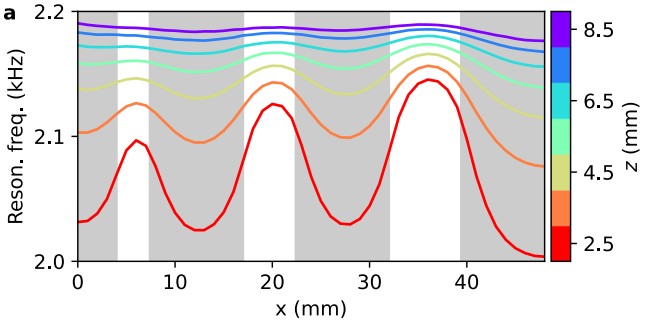

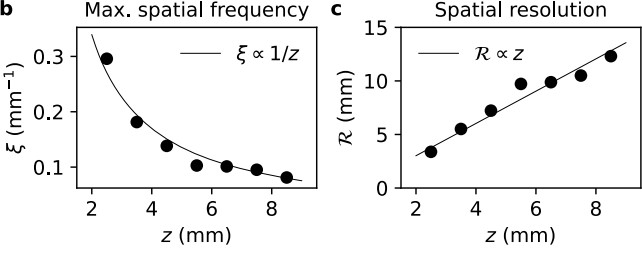

**Fig. 5 | Transverse resolution of the approach. a** Experimental measurements of the resonance frequency of the bubble for a 1D line scan above a resolution test sample composed of three lines (white bands in the figure) engraved in a stainless-steel plate (gray bands in the figure). The thickness of the plate is 8 mm, and engraved patterns go through the whole plate thickness. Performing these measurements for different distances $z$ ranging from 2.5 mm (red curve) to 8.5 mm (purple curve) demonstrates that the resolution rapidly degrades with the bubble-sample distance. **b** Maximum spatial frequency $\xi$ measured in these 1D line scans as a function of the bubble-sample distance $z$ (black points), along with a theoretical model predicting $\xi \propto 1/z$ (black curve). **c** Spatial resolution $\mathcal{R} = 1/\xi$ measured in these 1D line scans as a function of the bubble-sample distance $z$ (black points), along with the $\mathcal{R} \propto z$ prediction (black curve).

surrounding environment in a specific way. In our work, the contrast of reconstructed images is mostly driven by the frequency shift experienced by the bubble. This differs from analogous optical experiments implemented with fluorescent emitters[10–12], which are essentially sensitive to fluorescence lifetime variations and for which the frequency shift is too small to be detected. In analogy with these experiments, we could also in theory measure variations of the radiative linewidth[36], which is directly related to the local density of acoustic states[48,49]. However, the predominance of non-radiative damping in our experiments currently prevents us to measure radiative linewidth variations that are induced by the acoustic environment. Indeed, while a quality factor of around 74 is expected from the theoretical expression of the radiative linewidth $\Gamma_0 = 2\pi^2 f_0^2 d_0 / c_l$[19], we actually typically measure a quality factor of around 10 in our experiments. This difference clearly indicates the predominance of non-radiative damping, and further investigations will be necessary to identify the underlying sources of losses. Indeed, in addition to the contribution of the thermal boundary layer, we suspect that the presence of the cage also contributes to the non-radiative damping of the bubble, an effect that could potentially be mitigated with a different cage design.

To summarize, we introduced an approach to measure acoustic interactions in the near field of complex materials with a single resonating bubble, and we experimentally demonstrated super-resolution acoustic imaging of structured samples with a resolution two orders of magnitude smaller than the wavelength of the acoustic field. This approach can be used to probe acoustic properties of structured materials as well as soft tissues without mechanical contact, therefore providing interesting perspectives for acoustic microrheology[50–52]. Moreover, we highlight that the approach can be extended to the manipulation of several bubbles, opening up interesting perspectives for the study of multiple scattering and cooperative emission phenomena in complex acoustic environments and metamaterials[53–56].

## Methods

### Fabrication of the cages

Cubic cages are fabricated using a 3D printing technique based on digital light processing. A commercial resin (Monocure 3D, Grey Resin) is polymerized using a Photon Mono X (4K) Anycubic printer. Cubic cages with supports (see Fig. 1a) are designed using a computer-aided design software (FreeCad) and saved as STereoLithography files. These files are sliced in the vertical ($z$) direction, with a layer thickness of 0.05 mm. The printer is operated with a normal exposure time of 2 s, a bottom exposure time of 40 s, a $z$-lift distance of 6 mm, a $z$-lift speed of 3 mm/s, a $z$-retract speed of 3 mm/s, and a number of bottom layers (hooking at the printing platform) of six. Objects are fabricated layer by layer, with an upside-down orientation, resulting in a manufacturing time of around 40 min. After printing, the objects are rinsed in 2-propanol for 30 min. With a plateform area of 20 cm × 13 cm, twenty cages can be fabricated in one run. To ensure an efficient hydrophobicity of the structure, an additional water-repellent treatment (Glaco) is applied on the dried structures. The cages that we built have an external size of 3 mm, with a pillar thickness of 0.5 mm. As the water-air interfaces are typically located on the external faces of the cages[32], the volume of air trapped inside the cages is $V \simeq 20\ \text{mm}^3$, which yields an effective bubble size of $a_0 = V^{1/3} \simeq 2.7\ \text{mm}$ and an effective spherical bubble diameter of $d_{eq} = (6V/\pi)^{1/3} \simeq 3.4\ \text{mm}$.

### Experimental setup

Immersing a hydrophobic 3D-printed cage inside a tank filled with demineralized water (internal dimensions of the tank: 190 mm × 190 mm × 190 mm) directly leads to the formation of a bubble within the cage. The position of this caged bubble is then controlled using a 3D motorized stage (Newport ILS200PP). An arbitrary wavefront generator (Tiepie Handyscope HS5) is used to generate input electrical signals sampled at 500 kHz, at a repetition rate of 35 Hz, with a 14-bit

resolution and with a maximum amplitude of ±1.8 V. Each input signal is a Gaussian pulse, centered at 2 kHz and with a −3 dB bandwidth of 0.7 kHz. An underwater loudspeaker (Visaton FR 8 WP) fixed on the side of the tank converts this signal into an acoustic signal, which excites the bubble inside the tank. A transducer (Brüel & Kjær Miniature Hydrophone Type 8103) is used to convert the acoustic signal into an electrical signal, and also serves to hold the 3D-printed cage structure (see Fig. 1a). This signal is amplified by an amplifier (Brüel & Kjær Conditioning Amplifier Type 2692, bandpass filter 10 Hz/10 kHz, sensitivity 10 mV/Pa) before being transmitted to an USB oscilloscope (Tiepie Handyscope HS5), recording the measured signal during 25 ms at a sampling rate of 200 kHz with a 16-bit resolution.

### Description of the samples

To measure variations of the bubble dynamics close to a water-steel interface, we place a stainless-steel block (block dimensions: 100 mm × 100 mm × 30 mm) at the bottom of the tank. In the case of the water-air interface, we simply revert the orientation of the probe (see Fig. 2e), in order to be able to place it in the vicinity of the water-air interface (interface area: 190 mm × 190 mm).

The sample with the representation of the Eiffel tower (see Fig. 4c) is engraved in a stainless-steel plate of dimensions 153 mm × 98 mm × 8 mm. The fabrication process is based on water jet cutting with computer numerical control (CNC), with engraved patterns going through the whole plate thickness for this sample.

The sample with the acronym SNAM (see Fig. 4f) is engraved in a stainless-steel plate of dimensions 159 mm × 73 mm × 8 mm. By CNC drilling, we engrave patterns over a thickness of 7 mm, leaving a thin continuous layer of steel at the bottom of the plate (this ensures that the central part of the letter "A" remains linked to the other part of the structure). We then cover this letter with adhesive tape (approximate thickness of 30 μm), preventing water to infiltrate in the engraved area.

The sample with three lines (see Fig. 5a) is engraved in a stainless-steel plate of dimensions 70 mm × 65 mm × 8 mm. By CNC drilling, we engrave three lines of equal height (42 mm) and of width equal to 3 mm, 5 mm, and 7 mm, respectively. Engraved patterns go through the whole plate thickness for this sample.

### Acquisition procedure

The 1D line scans shown in Fig. 2d, h are composed of 61 equally-spaced measurement points. For these measurements, the step size is 0.5 mm, and each data point is obtained by averaging over 200 measurements to reduce the influence of noise fluctuations. The cage is first placed in contact with the interface, and the first data point is taken after moving the bubble 1 mm away from the interface. Considering that the length of the cage side is 3 mm, the first data point is thus taken at a distance $z = 2.5$ mm to the interface (the distance is defined from the interface to the estimated position of the center of the bubble). Note that, due to the low signal-to-noise ratio of the first data point in the case of the water-air interface, we removed this point and show data only starting from $z = 3$ mm.

The images of the Eiffel tower (Fig. 4d, e) are composed of 33 × 61 measurements points, and the images of the SNAM acronym (Fig. 4g, h) are composed of 69 × 27 measurement points. In both cases, the step size is 2 mm, and each data point is obtained by averaging over 20 measurements. The cage is first placed in contact with the sample, and then retracted by 1 mm before scanning the probe in the transverse plane; the center of the bubble is thus at a distance $z = 2.5$ mm from the interface. Raster scans are then performed with the fast scanning direction along the longest dimension of the sample under study (the vertical dimension for the Eiffel tower, and the horizontal dimension for the SNAM acronym).

The 1D line scans shown in Fig. 5a are composed of 49 equally-spaced measurement points. For these measurements, the step size is 1 mm, and at each position we perform 200 measurements of the same

field, in order to obtain different noise realizations as needed for the FRC-inspired analysis. The cage is first placed in contact with the interface, and the first line scan is performed after retracting the cage by 1 mm, which corresponds to a distance $z = 2.5$ mm. We then repeat this procedure for different values of $z$, ranging from $z = 2.5$ mm to $z = 8.5$ mm.

In all experiments, we first measure the field at each position in the presence of the bubble $\phi_w(\mathbf{r}, t)$. This is simply achieved by immersing a dry cage into the water tank; this cage then naturally confines and stabilizes an air bubble. Once the scan with the caged air bubble is finished, we remove the air from the cage by injecting water inside the cage using a pipette. This allows us to measure the field at each position in the absence of the bubble $\phi_{w/o}(\mathbf{r}, t)$.

## Data processing

From measurements of the scattered field in the time domain with the bubble $\phi_w(\mathbf{r}, t)$ and without the bubble $\phi_{w/o}(\mathbf{r}, t)$, we calculate the normalized scattering amplitude

$$A_s(\mathbf{r}, \omega) = \frac{\hat{\phi}_w(\mathbf{r}, \omega) - \hat{\phi}_{w/o}(\mathbf{r}, \omega)}{\hat{\phi}_{w/o}(\mathbf{r}, \omega)} , \qquad (4)$$

where $\hat{\phi}_w(\mathbf{r}, \omega)$ and $\hat{\phi}_{w/o}(\mathbf{r}, \omega)$ denote the Fourier transforms of $\phi_w(\mathbf{r}, t)$ and $\phi_{w/o}(\mathbf{r}, t)$, respectively. In order to extract the central frequency of the resonance from the normalized power spectral density $|A_s(\mathbf{r}, \omega)|^2$, we use a Lorentzian function

$$f(\omega) = \frac{K}{2\pi} \frac{\gamma}{(\omega - \omega_0)^2 + (\gamma/2)^2} , \qquad (5)$$

where $K$ is a scaling factor, $\omega_0 = 2\pi f_0$ is the angular resonance frequency, and $\gamma$ is the linewidth. These three parameters are the free parameters of the fitting procedure. We first identify the frequency associated with the maximum value of $|A_s(\mathbf{r}, \omega)|^2$, and we then fit a Lorentzian function to the data on a restricted frequency window around this maximum (we set the width of this window to 0.6 kHz). The influence of the number of measurements per data point on the measured resonance frequency is discussed in Supplementary Materials, Section 5.

## FDTD simulations

Numerical simulations were implemented with a finite-difference time-domain solver of the elastodynamic equations, with a freely available software developed in our group[39]. We followed an approach similar to the one implemented in our earlier works[32,34]. In particular, we used a Cartesian mesh with a spatial step around 150 μm (see details in Supplementary Materials, Section S2), within a total simulation volume $20 \times 20 \times 20$ mm$^3$. Water and air were modeled as non-dissipative fluids, and the cage itself was neglected. The simulation volume was either surrounded by perfectly matched layers (PML) to mimic an unbounded medium, or by stress-free BCs to mimic the tank boundaries. Wideband pressure pulses in the kHz range (2 kHz center frequency, 100% −6 dB relative bandwidth) were propagated with and without the presence of the bubbles, analogous to the experimental situation, to derive the resonant frequency of the bubble from the pressure signals. To assess the sensitivity of the predicted resonant frequency to the various simulation parameters (including the spatial grid step, the total simulation volume as compared to the volume of the bubble, thickness of the PML, etc...), several simulations were run by varying these parameters. The accuracy on the resonant frequency was estimated to be smaller than 1% relative error.

## Data availability

The data generated in this study and Python scripts for data processing have been deposited in the Data Repository Grenoble Alpes database with the following https://doi.org/10.57745/XNJB5K.

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

## Acknowledgements

The authors thank Irène Wang for insightful discussions, Philippe Moreau for technical support, and Bruno Peccoud for the design of Fig. 1a. This project has received funding from the French Agence Nationale de la Recherche (ANR-23-CE42-0016, E.B.).

## Author contributions

E.B. proposed the use of a caged bubble as a resonant probe for near-field acoustic imaging; D.B. and E.B. designed the experiments. D.B. performed the experiments and processed the data. D.B. and B.D. compared the experimental results to the theory. E.B. performed and analyzed the FDTD simulations. O.S. and P.M. designed and fabricated the caged bubbles. All authors discussed and analyzed the results. D.B. wrote the first draft of the manuscript, and all authors contributed to the writing and finalization of the manuscript.

## Competing interests

D.B., B.D., P.M., and E.B. filed a pending patent entitled "Scanning near-field acoustic microscopy using resonant gas bubble probe". O.S. declares no competing interests.
