## [Transparent Peer Review file · Nature Communications]

Near-field acoustic imaging with a caged bubble

Corresponding Author: Professor Emmanuel Bossy

Version 0:

Reviewer comments:

Reviewer #1

(Remarks to the Author)

General Comments:

The manuscript presents a novel study on the application of Scanning Near-field Acoustic Microscopy (SNAM) in understanding volcanic eruptions and water circulation in vascular plants. The research is well-motivated and addresses important scientific questions. However, there are several areas where the manuscript could be strengthened. Clarification of the practical applications of SNAM, deeper integration of the methods with the study's main objectives, and additional details on experimental setups and methodologies are needed to enhance the clarity and impact of the work. Furthermore, the comparison with existing techniques and a more thorough discussion of the imaging parameters would provide valuable context and support for the study's findings. The following detailed comments and suggestions aim to help the authors improve the manuscript's coherence and comprehensiveness.

Specific Comments:

Line 5: In the abstract, the authors claim that this study is important for the understanding of volcanic eruptions and water circulation in vascular plants. This is a significant claim. Could the authors provide more context or specific examples of how their findings contribute to these fields in the main texts of the manuscript?

Line 15: At the end of the abstract, it would be beneficial for the authors to highlight the potential applications of using Scanning Near-field Acoustic Microscopy (SNAM) in both industry and clinical settings. Currently, the manuscript lacks a clear outline of these applications, which would enhance the reader's understanding of the practical significance of the study.

Line 24: The authors assert that acoustic waves, rather than light, are used to specifically assess the elastic properties of materials. While this claim is accurate, its direct relevance to the main topic of this study is not clearly established. Could the authors elaborate on how this sentence integrates with the overall research objectives?

Line 24: The authors describe microbubble-based ultrasound localization microscopy as the acoustic analogue of photo-activated localization microscopy (PALM). However, in photo-activated localization microscopy, laser pulses are used to activate and deactivate fluorescence within cells. Please also reference the technique 'acoustic wave sparsely activated localization microscopy', which uses the activation and deactivation of nanodroplet contrast agents to achieve the super-resolution in ultrasound imaging. This could be closer to the acoustic analogue of PALM. Relevant references include:

1. Ge Zhang et al., "Acoustic wave sparsely activated localization microscopy (AWSALM): Super-resolution ultrasound imaging using acoustic activation and deactivation of nanodroplets." *Appl. Phys. Lett.* 113, 014101 (2018). <https://doi.org/10.1063/1.5029874>
2. G. Zhang et al., "Fast Acoustic Wave Sparsely Activated Localization Microscopy: Ultrasound Super-Resolution Using Plane-Wave Activation of Nanodroplets," *IEEE Trans. Ultrason. Ferroelectr. Freq. Control*, vol. 66, no. 6, pp. 1039-1046, June 2019. <https://doi.org/10.1109/TUFFC.2019.2906496>

Line 42: Regarding Figure 1, the bubble size is stated as $\lambda/280$. Is the bubble size variable with λ in this study? If not, it would be helpful for the authors to provide the exact value of the bubble size for clarity.

Line 42: For Figure 1B, the experimental setup appears quite complex, involving a large water tank, speaker, transducer, caged bubble, imaging target immersed in water, and a 3D stage motor. How feasible is this setup for real-world applications? Could the authors discuss potential adaptations or simplifications for practical use?

Line 152: In Figure 4, the manuscript demonstrates super-resolution near-field imaging with a single resonating bubble. A key point of interest is how this method compares with acoustic microscopy using higher frequencies. Can the sample be imaged using higher frequency ultrasound? What are the limitations or benefits of using lower frequencies in this context? If higher frequency imaging is viable, why is lower frequency imaging preferred or necessary?

Line 271: For the image of the Eiffel Tower composed of 33 x 61 measurement points, how did the authors determine this specific matrix of measurement points? Could this matrix be expanded to achieve a finer grain size or pixel size for the final image?

Line 273: The authors mention that each data point result is an average of 20 measurements. How long does each individual measurement take? Is it necessary to average over 20 measurements to obtain reliable results, or could fewer measurements suffice?

Reviewer #2

(Remarks to the Author)

1. The paper is mostly about a SNAM device, not really on the sound of a cubic bubble. A more right-on-the-target title (and the theme) of the paper would be better.
2. The first sentence of the abstract, "The study of gas bubbles in liquid ...", seems redundant.
3. There exists a long list of literature on spherical bubbles, and thus scattering from a cubic bubble deserves some close investigation. However, the only detailed discussion on a cubic bubble is just one figure, Fig. 3, which shows no difference in comparison with a spherical bubble. In this part, the authors also heavily rely on the results from a numerical simulation tool and the idealized potential theory result of Morioka in 1974. (Not sure why the numerical simulation tool is described as an FDTD "resolution" in Line 103.)
4. In Fig. 3C, the perfect match seems because the effective diameter was treated "as a free parameter". First, what does "a free parameter" mean here? A clear definition or how it is determined is needed. Second, if "a free parameter" means that it can be adjusted to warrant a perfect match, then what does the perfect match of the results mean?
5. The rest of the paper is describing the technicality of how the device works. These discussions can be applied to a device with a spherical bubble as well, not particularly only for a cubic bubble.

Version 1:

Reviewer comments:

Reviewer #1

(Remarks to the Author)

I have reviewed the revised manuscript and am pleased to see that the authors have thoroughly addressed all of my previous comments and suggestions. The revisions have significantly improved the clarity and quality of the work, and I am satisfied with the changes made. Therefore, I would like to recommend this revised version for publication, as it now meets the necessary standards for acceptance.

Reviewer #2

(Remarks to the Author)

The authors have diligently revised the manuscript, resulting in a very much improved paper. I have the following comments on the revision:

1. The procedure of fitting has been discussed in detail with an additional section in Method for that. However, the main text still only mentions one fitting parameter, the effective diameter. Please also clarify this in the main text so that the readers won't be confused.
2. In the description of the fitting procedure, the authors indicate that they used 100 terms for Morioka's expression. Please explain the reason why 100 terms are selected, instead of other numbers of terms.
3. It seems the fitting procedure used to determine the two fitting parameters is the least square fitting. Please provide the number and range of samples and the uncertainty levels of the fitting.

Manuscript NCOMMS-24-36174: Reply to Referees

We thank the reviewers for their time and their feedback on our manuscript. Please find below our point-by-point responses to the reviewers' comments.

Response to Reviewer #1

Reviewer #1 states:

“The manuscript presents a novel study on the application of Scanning Near-field Acoustic Microscopy (SNAM) in understanding volcanic eruptions and water circulation in vascular plants. The research is well-motivated and addresses important scientific questions. However, there are several areas where the manuscript could be strengthened. Clarification of the practical applications of SNAM, deeper integration of the methods with the study’s main objectives, and additional details on experimental setups and methodologies are needed to enhance the clarity and impact of the work. Furthermore, the comparison with existing techniques and a more thorough discussion of the imaging parameters would provide valuable context and support for the study’s findings. The following detailed comments and suggestions aim to help the authors improve the manuscript’s coherence and comprehensiveness.”

Our response:

We thank the reviewer for his/her report on our work. We have built on the specific comments and suggestions of the reviewer to improve the clarity of our manuscript and avoid any misunderstanding. Please find below our detailed answer to these comments.

Reviewer #1 states:

“Line 5: In the abstract, the authors claim that this study is important for the understanding of volcanic eruptions and water circulation in vascular plants. This is a significant claim. Could the authors provide more context or specific examples of how their findings contribute to these fields in the main texts of the manuscript?”

Our response:

The objective of our introductory sentence is to emphasize the importance of studying the formation of bubbles and their dynamics in various research applications. Given the multidisciplinary audience of Nature Communications, it seems important to us to give the reader some general context about bubbles, which are the major physical components of our approach. However, we never claimed that our study was important to specifically study volcanic eruptions and water circulation in water plants.

Our study is important, first, because it enables one to easily measure the dynamics of bubbles in complex environments, which is of strong interest for fundamental studies on multiple scattering effects and collective emission phenomenon (as mentioned in lines 223–225 of our manuscript, see also Refs. [53–56]). Moreover, from an application point of view, our approach paves the way for the development of low-cost acoustic microscopes based on a very cheap probe: an air bubble in water (see lines 181–202 of our manuscript; see also below our answer to the next reviewer’s comment).

Changes to the manuscript: To avoid any confusion, we have reformulated the first sentence of the abstract: “Gas bubbles are ubiquitous in many research applications ranging from ultrasound

imaging and drug delivery to the understanding of volcanic eruptions and water circulation in vascular plants.” (lines 4–5).

Reviewer #1 states:

“Line 15: At the end of the abstract, it would be beneficial for the authors to highlight the potential applications of using Scanning Near-field Acoustic Microscopy (SNAM) in both industry and clinical settings. Currently, the manuscript lacks a clear outline of these applications, which would enhance the reader’s understanding of the practical significance of the study.”

Our response:

As mentioned in our response above, a major potential application of our work is the development of low-cost acoustic microscopes. In the revised version of the manuscript, we provide more details on this potential application.

Changes to the manuscript: We have added a sentence at the end of the abstract to highlight a major potential application of our approach: “As a major potential application, our approach paves the way for the development of low-cost acoustic microscopes based on caged bubbles.” (lines 15–16)

To provide a better outline of possible applications of our technique, we also added the following paragraph in the text: “On the one hand, scaling up the size of the bubble cage in the centimeter range [34] would enable locally-resolved rheological measurements of interfaces in the few hundred Hz frequency range, otherwise impossible with diffraction-limited transducers which would be several meters in size. On the other hand, our approach could be implemented on the micrometer scale by engineering cages via 3D microfabrication [46], which would allow to reach a micrometer resolution using conventional MHz electronics and transducers. Current commercial acoustic microscopes, widely used for non-destructive testing in the industry and biomedical applications on the micrometer scale [14,47], are based on expensive electronics and transducers working in the GHz range. Our approach would enable acoustic microscopy with micrometer resolution at the cost of a MHz ultrasound imaging system, that are typically one or two orders of magnitude cheaper than GHz systems.” (lines 185–194)

Reviewer #1 states:

“Line 24: The authors assert that acoustic waves, rather than light, are used to specifically assess the elastic properties of materials. While this claim is accurate, its direct relevance to the main topic of this study is not clearly established. Could the authors elaborate on how this sentence integrates with the overall research objectives?”

Our response:

The overall research objective is to develop a novel type of near-field acoustic imaging system. The purpose of this sentence is to insist on the fact that scanning near-field optical microscopy, which is a well-known approach, is limited to measuring optical properties. The purpose of developing a scanning near-field acoustic microscopy system is to provide complementary information on mechanical properties.

Changes to the manuscript: We clarified the reason why we used acoustic waves in our study: “For many applications in nondestructive testing and biological imaging [13,14], it is necessary to be specifically sensitive to the elastic properties of materials. In this context, it is thus relevant to use acoustic waves instead of light as a sensing mechanism.” (lines 24–26)

Reviewer #1 states:

“Line 24: The authors describe microbubble-based ultrasound localization microscopy as the acoustic analogue of photo-activated localization microscopy (PALM). However, in photo-activated localization microscopy, laser pulses are used to activate and deactivate fluorescence within cells. Please also reference the technique ‘acoustic wave sparsely activated localization microscopy’, which uses the activation and deactivation of nanodroplet contrast agents to achieve the super-resolution in ultrasound imaging. This could be closer to the acoustic analogue of PALM. Relevant references include:

*1. Ge Zhang et al., "Acoustic wave sparsely activated localization microscopy (AWSALM): Super-resolution ultrasound imaging using acoustic activation and deactivation of nanodroplets." Appl. Phys. Lett. 113, 014101 (2018).
<https://doi.org/10.1063/1.5029874>*

*2. G. Zhang et al., "Fast Acoustic Wave Sparsely Activated Localization Microscopy: Ultrasound Super-Resolution Using Plane-Wave Activation of Nanodroplets," IEEE Trans. Ultrason. Ferroelectr. Freq. Control, vol. 66, no. 6, pp. 1039-1046, June 2019.
<https://doi.org/10.1109/TUFFC.2019.2906496>”*

Our response:

We agree that “acoustic wave sparsely activated localization microscopy” is a closer analogue to PALM than ultrasound localization microscopy.

Changes to the manuscript: We now cite “Appl. Phys. Lett. 113, 014101 (2018)” as Ref. [23] of the revised manuscript.

Reviewer #1 states:

“Line 42: Regarding Figure 1, the bubble size is stated as $\lambda/280$. Is the bubble size variable with λ in this study? If not, it would be helpful for the authors to provide the exact value of the bubble size for clarity.”

Our response:

With this label, we wanted to insist on the very large ratio between the two scales. However, we agree that it could lead the reader to think that we varied the bubble size, which was indeed not the case.

Changes to the manuscript: This has now be clarified by stating explicitly the bubble size (2.7 mm) and the wavelength (75 cm) in Figure 1a. We also clarified how this size is defined in the Methods (lines 238–241).

Reviewer #1 states:

“Line 42: For Figure 1B, the experimental setup appears quite complex, involving a large water tank, speaker, transducer, caged bubble, imaging target immersed in water, and a 3D stage motor. How feasible is this setup for real-world applications? Could the authors discuss potential adaptations or simplifications for practical use?”

Our response:

We would first like to highlight that our system is no more complex than any optical or acoustic scanning near-field microscopes (which all include motorized translation stages), but on the

contrary benefits from the simplest possible imaging tip in the form of a caged bubble.

The size of the water tank is mostly conditioned by the size of the sample. Here it was several tens of centimeters wide because we provided a proof-of-concept experiment in the kHz range, and thus relevant sizes for test samples is of several centimeters. But as for the approach itself, the size of the setup scales with the size of the sample, which could thus be more compact for acoustic microscopy applications.

Finally, in the kHz range, it was easier and more practical for our proof-of-concept experiment to use a loudspeaker and a hydrophone, but systems may be more compact by use of reciprocal pulse-echo transducer.

Changes to the manuscript: We added a paragraph that now emphasizes how to make the setup more compact: “Note that a standard focused single-element MHz transducer could be used to both excite and detect sound from the bubble probe, which would result in a more compact setup as compared to our proof-of-concept setup with a hydrophone and a speaker.” (lines 200–202).

Reviewer #1 states:

“Line 152: In Figure 4, the manuscript demonstrates super-resolution near-field imaging with a single resonating bubble. A key point of interest is how this method compares with acoustic microscopy using higher frequencies. Can the sample be imaged using higher frequency ultrasound? What are the limitations or benefits of using lower frequencies in this context? If higher frequency imaging is viable, why is lower frequency imaging preferred or necessary?”

Our response:

This is indeed an important point. To get acoustic images with millimetric resolution, a conventional approach using acoustic microscopy with a MHz single-element focused pulse-echo transducer would work similarly well here in terms of resolution. There are, however, two reasons why our approach is interesting and useful from a practical point of view.

First, our approach provides several contrast mechanisms (as shown e.g. in Figure 4 G and H), which are also different from those obtained from MHz ultrasound. This feature could be very useful to recover elastic properties of materials from ultrasound measurements.

Second, and more importantly, the purpose of our work is to introduce a completely scalable method, whose principle can apply to a very wide frequency range. Our proof-of-concept experiment generally demonstrates that, using a caged bubble, we can work with a frequency about a thousand times lower than what is required by conventional approaches to get the same resolution. For simplicity, we did it in the kHz frequency range (i.e. mm resolution), but because the approach is inherently scalable, our demonstration paves the way for the development of devices with many different bubble sizes. Indeed, we expect that, in the future, our approach will demonstrate its full application potential with both large (centimetric) and small (micrometric) bubbles size. Let us take the example of acoustic microscopy with micrometer resolution: it is doable and exists, but at the cost of complex and expensive GHz transducers and GHz electronics (typically one million euros). Our technique will then come as a preferred one based on cost and simplicity arguments, as MHz ultrasound devices are one or two orders of magnitude cheaper than GHz ultrasound devices. Considering another example in a lower frequency range (in the tens of hundred Hz range), our approach will deliver cm-resolved images. In this context, our approach will be both preferred and necessary, as focused transducers in this range would be too large for practical use (they would be several meters in size).

Changes to the manuscript: In the revised manuscript, we highlighted that we introduce in our paper a general approach that is scalable, with potential applications for both low-frequency and high-frequency ultrasound imaging: “On the one hand, scaling up the size of the bubble cage in the centimeter range [34] would enable locally-resolved rheological measurements of interfaces in the few hundred Hz frequency range, otherwise impossible with diffraction-limited transducers which would be several meters in size. On the other hand, our approach could be implemented on the micrometer scale by engineering cages via 3D microfabrication [46], which would allow to reach a micrometer resolution using conventional MHz electronics and transducers. Current commercial acoustic microscopes, widely used for non-destructive testing in the industry and biomedical applications on the micrometer scale [14,47], are based on expensive electronics and transducers working in the GHz range. Our approach would enable acoustic microscopy with micrometer resolution at the cost of a MHz ultrasound imaging system, that are typically one or two orders of magnitude cheaper than GHz systems.” (lines 185–194)

Reviewer #1 states:

“Line 271: For the image of the Eiffel Tower composed of 33×61 measurement points, how did the authors determine this specific matrix of measurement points? Could this matrix be expanded to achieve a finer grain size or pixel size for the final image?”

Our response:

From a practical perspective, taking a step size of 1 mm instead of 2 mm would lead to a $\times 4$ increase of the acquisition time, which would be 8 hours instead of 2 hours. This would be rather long, but still manageable experimentally as our caged bubbles are remarkably stable in time (see Supplementary Figure S1).

However, conceptually, we would not gain anything by using a finer step size. Indeed, we carefully chose our step size (2 mm) in order to fully demonstrate the resolving power of our method (around 3 mm). We could of course choose a finer step size but, since the resolution of our approach is limited by the bubble size to approximately 3 mm, we would not get any additional useful information.

Changes to the manuscript: In the supplementary information of our revised manuscript, we now mention the maximum acquisition time allowed by our method: “Overall, this strategy allows performing long acquisitions with the same bubble (up to dozens of hours).”

Reviewer #1 states:

“Line 273: The authors mention that each data point result is an average of 20 measurements. How long does each individual measurement take? Is it necessary to average over 20 measurements to obtain reliable results, or could fewer measurements suffice?”

Our response:

As mentioned in the Methods section, we operate our pulse generator at a repetition rate of 35 Hz. Therefore, each individual measurement takes approximately 29 ms.

It is in fact possible to rely on one single measurement per data point. As an example, we reproduced Figure 5A of the manuscript, averaging each data point over either 1, 20, or 200 measurements. We obtained the results presented in the figure below, on which it appears that relying on a single measurement is definitely possible, at the cost of reducing the signal-to-noise ratio.

Figure 1: Measurements of the resonance frequency of the bubble for the 1D line scans shown in Fig. 5a of the manuscript. The number of measurements used at each point is either 1 (a), 20 (b) or 200 (c).

In practice, in our imaging experiments, each data point was obtained by averaging over 20 measurements; even though this was not strictly necessary, it gave us a better signal-to-noise ratio without strongly affecting the total acquisition time. Indeed, taking the example of the Eiffel tower sample, the total acquisition time was 1 hour and 58 min but, during this time, only around 19 min were spent for actually measuring the data. This clearly shows that the total acquisition time is mostly due to the time spent by the motorized stage to go from position to position, and not by the measurement time. For instance, if instead of averaging over 20 measurements, we had performed this experiment by taking 1 measurement at each point, the measurement time would have been of around 1 min, but the total acquisition time would have remain rather long (around 1 hour and 39 min) due to the time that the motorized stage takes during its motion.

Changes to the manuscript: We included these results and the discussion in the Supplementary Materials, Section S5.

Response to Reviewer #2

Reviewer #2 states:

“1. The paper is mostly about a SNAM device, not really on the sound of a cubic bubble. A more right-on-the-target title (and the theme) of the paper would be better.”

Our response:

We thank the reviewer for his feedback on the title, which is a point that we had discussed at length between co-authors. By using cubic, we wanted to emphasize the fact that we exert a large degree of control over the bubble, rather than emphasizing its exact shape. Our paper is indeed about a SNAM device, whose novelty comes from the nature of its tip (a caged bubble) which confers its strongly subwavelength resolution.

Changes to the manuscript: According to the reviewer’s suggestion, we changed the title to “Near-field acoustic imaging with a caged bubble”, which carries our intended message in a more direct way.

Reviewer #2 states:

“2. The first sentence of the abstract, “The study of gas bubbles in liquid . . .”, seems redundant.”

Our response:

We agree with the reviewer that bubbles are usually defined as balls of gas that appears in a liquid. However, the word “bubble” is also sometimes used to refer to a ball of gas surrounded by liquid that floats in the air, a situation that does not correspond to our experiment.

Changes to the manuscript: Given the context of our work, we choose to follow the reviewer’s suggestion and formulate the first sentence of the abstract as follows: “Bubble are ubiquitous in many research applications [...]” (line 4).

Reviewer #2 states:

“3. There exists a long list of literature on spherical bubbles, and thus scattering from a cubic bubble deserves some close investigation. However, the only detailed discussion on a cubic bubble is just one figure, Fig. 3, which shows no difference in comparison with a spherical bubble. In this part, the authors also heavily rely on the results from a numerical simulation tool and the idealized potential theory result of Morioka in 1974. (Not sure why the numerical simulation tool is described as an FDTD “resolution” in Line 103.)”

Our response:

We agree with the reviewer that scattering from a cubic bubble deserves some close investigation. In fact, our team is closely investigating the scattering properties of cubic bubbles since 2018. Let us discuss these properties in two different situations, depending on the environment of the bubble.

1) It is first interesting to understand the behavior of a cubic bubble in a free-space environment. This was studied in depth in some of our earlier works, where we first introduced cubic bubbles (Ref. [32]) before studying bubbles of polyhedral shapes (Ref. [34]). The conclusion of these studies is that the resonance frequency of a bubble in a free-space environment is dictated almost entirely by its volume, up to one or two percent accuracy (see Ref. [34]).

2) In the context of the present work, we now demonstrate that the resonance of a cubic bubble close to an interface behaves very similarly to that of spherical bubbles, which was far from being evident in the first place. More precisely, we demonstrate that the way a cubic bubble resonates in the vicinity of hard or soft interfaces is nearly the same for a cubic bubble and for a spherical one. The main result is presented in Fig 3: this figure not only shows a first experimental demonstration of the performances of our SNAM device, but also reports a first-of-its-kind experimental validation of the Morioka predictions with a bubble of controlled position. With this experiment, we indeed validate the predictions of Morioka’s analytical theory and, at the same time, we also demonstrates that the shape of the bubble itself has almost no influence. Because this influence is so small, we did not discuss it more in the main text, but we do provide a thorough analysis in the Supplementary Material, Sections S2.1 and S2.2. There, we study the small but still significant difference that can be revealed by accurate FDTD simulations between cubic and spherical bubbles. The additional information brought here by the simulations is thus essentially about the tiny difference between cube and sphere of the same volume, as well as an additional validation of Morioka’s theory for any bubble shape.

We thus had put a lot of efforts in investigating the scattering properties of cubic bubbles (both in earlier works and in the present work), and the result is essentially that there are no strong differences between a cubic bubble and a spherical bubble of the same volume. Consequently, as noted by the reviewer in his last comment, our proposed method is quite independent on the shape of the bubble, it just needs that the bubble resonates (this is mentioned in lines 63–65 of the manuscript). The use of the term “resolution” is indeed a mistake, thanks for pointing this

out; we confused the words solving and resolution, which are both translated by the same word in French.

Changes to the manuscript: To highlight that we investigated the scattering properties of cubic bubbles in free-space environment in some of our earlier works, we have added the following sentence: “The resonance frequency of a cubic bubble of side length a_0 is essentially driven by the volume V of the gas in the bubble, as previously investigated using bubbles of polyhedral shapes [34].” (lines 65–66).

To highlight the new information given in the present work regarding the comparison between spherical and cubic bubbles, we also added the following sentence: “this also demonstrates that the resonance frequency of a cubic bubble close to an interface behaves very similarly to that of a spherical one” (lines 117–118).

We corrected our mistake regarding the words “solving” and “resolution” in the sentence “we conducted 3D numerical simulations based on a finite-difference time-domain (FDTD) solver of the elastodynamic equations.” (line 104).

Reviewer #2 states:

“4. In Fig. 3C, the perfect match seems because the effective diameter was treated “as a free parameter”. First, what does “a free parameter” mean here? A clear definition or how it is determined is needed. Second, if “a free parameter” means that it can be adjusted to warrant a perfect match, then what does the perfect match of the results mean?”

Our response:

We agree with the reviewer that one must be careful when comparing experimental results to theoretical predictions while fitting some parameters to the data. In the following, we demonstrate that the observed values of these parameters are consistent with our a priori knowledge and, more importantly, that they have no effect on the shape of the model function (these parameters only amount to translate the model function).

First, let us recall what is the model function:

$$f_{\pm}(\Delta z, d_{\text{eq}}, z_0) = \left(\frac{c_g \sqrt{3\rho_g/\rho_l}}{\pi d_{\text{eq}}} \right) \times \sqrt{\sum_{n=0}^{+\infty} \frac{(\mp 1)^n \sinh(\beta)}{\sinh[(n+1)\beta]}} \quad (1)$$

where $\cosh(\beta) = 2(z_0 + \Delta z)/d_{\text{eq}}$. In this function, the first term (inside braces) corresponds to Minnaert’s formula giving the free-space resonance frequency, while the second term (under the square root) corresponds to Morioka’s result giving the distance dependence (note that in practice we truncate the series to $n = 100$). Taking $c_g = 340$ m/s, $\rho_g = 1.2$ kg/m³ and $\rho_l = 1000$ kg/m³, we are left with three parameters: the effective bubble diameter d_{eq} , the bubble-interface distance for the first measurement point z_0 (when the bubble is closest to the interface) and the relative displacement applied by the motorized stage for all other measurement points Δz . While Δz is precisely controlled by the stage in the experiment, it is difficult to precisely control the value of two other parameters d_{eq} and z_0 . d_{eq} could be accessible by either precisely measuring the volume of the bubble (e.g. with optical measurements), or by measuring the free-space resonance frequency of the bubble (our tank is however too small to do so). z_0 could be accessible by precisely measuring the distance-to-interface distance (using e.g. optical measurements). Precise optical measurements of d_{eq} and z_0 are, however, not easily achievable with our setup, and it is not a central aspect of our work (which is essentially about introducing an new SNAM device).

Thus, we opted for another strategy: determining experimentally these parameters directly from the data, by treating d_{eq} and z_0 as free parameters. They are numerically determined by solving

$$d_{\text{eq}}, z_0 = \underset{d', z'}{\operatorname{argmin}} \left(\sum_i [f_{\pm}^{\text{exp}}(\Delta z_i) - f_{\pm}(\Delta z_i, d', z')]^2 \right), \quad (2)$$

where $f_{\pm}^{\text{exp}}(\Delta z_i)$ denotes the experimental results of the 1D line scans in the z -direction. We are confident that this procedure fully makes sense here:

- In the case of the water-steel interface, we obtain an effective diameter of 3.1 mm and a bubble-interface distance z_0 of 2.3 mm. In the case of the water-air interface, we obtain an effective diameter of 3.4 mm and a bubble-interface distance z_0 of 2.6 mm. These values are consistent with what can be expected from the geometry of the cages (volume of trapped air $V \leq 20 \text{ mm}^3$, which gives $d_{\text{eq}} \leq 3.4 \text{ mm}$) and from our procedure to control the minimal bubble-interface distance (we establish the contact between the cage and the interface by eye and then retract the cage by 1 mm, which gives a distance between the interface and the center of the bubble of $z_0 \simeq 2.5 \text{ mm}$).
- In practice, varying z_0 amounts to horizontally translate the function f_{\pm} , and varying d_{eq} amounts to vertically translate this function (up to a very good approximation). This is evidenced in the figure below, in which we varied the value of the free parameters around their true values in order to demonstrate their effect upon the shape of the model function. The precise value of these parameters has no effect on the shape of the model function. The very good match between experimental results and the theoretical model thus means that the shape of the distance dependence that we experimentally observe is very well described by the model.

Figure 2: Resonance frequency of the bubble as a function of the distance to the interface, along with theoretical predictions. The value of the free parameter d_{eq} is varied in the left figure (vertical translation of the curve) and the value of the free parameter z_0 is varied in the right figure (horizontal translation of the curve).

- Finally, we can mention that we have validated the theoretical result of Morioka also with numerical simulations that closely model our experiment, and in this case no free parameter was involved (see Supplementary Materials, Section 2).

Overall, the fact that the shape of the measured distance dependence is well described by the model provides a first-of-its-kind experimental validation of the Morioka predictions with a bubble of controlled position, something that is only made possible by the use of a stable caged bubble that can be accurately manipulated. In addition, it demonstrates that a theory derived for a spherical bubble works and fit nearly perfectly for a cubic bubble.

Changes to the manuscript: We added a section in the Methods (“Fitting procedure”) to clarify how the free parameters are defined and to explain why the results of the fit are meaningful (lines 308–320).

Reviewer #2 states:

“5. The rest of the paper is describing the technicality of how the device works. These discussions can be applied to a device with a spherical bubble as well, not particularly only for a cubic bubble.”

Our response:

We fully agree with the reviewer: we modified the title of the paper to avoid any misleading conclusion that the shape would be important. What is key to our work however is that a caged bubble is needed. It turns out that cubic bubbles/cages are the easier ones to manufacture, and spherical cages are the most complex ones to manufacture.

Changes to the manuscript: We have changed the title of the manuscript to avoid any misleading conclusion.

Concluding words

We thank again all of our two Reviewers for their helpful input, and we hope that our updated manuscript can now be recommended for publication in Nature Communications.

Manuscript NCOMMS-24-36174: Reply to Referees

We thank the reviewers for their time and their feedback on our manuscript. Reviewer #1 now recommends this version of the manuscript for publication. Please find below the answer to the questions of Reviewer #2.

Reviewer #2 states:

“1. The procedure of fitting has been discussed in detail with an additional section in Method for that. However, the main text still only mentions one fitting parameter, the effective diameter. Please also clarify this in the main text so that the readers won't be confused.”

Our answer

We thank the reviewer for pointing this out. In the main text, we added the following sentence: "In the experiments, both the effective bubble diameter and the minimal bubble-interface distance need to be treated as free parameters (see Supplementary Materials, Section 6)."

Reviewer #2 states:

“2. In the description of the fitting procedure, the authors indicate that they used 100 terms for Morioka's expression. Please explain the reason why 100 terms are selected, instead of other numbers of terms.”

Our answer

To numerically evaluate any infinite series, it has to be truncated. Here, by comparing the result obtained with 100 terms and 1000 terms, we observed no difference up to machine precision, indicating that 100 terms are sufficient to correctly estimate the series. In the Supplementary Information (Section S6), we added the following sentence: "Note that in practice we need to truncate the series for their numerical evaluation; here we keep the first 100 terms, which is sufficient to estimate the series up to machine precision."

Reviewer #2 states:

“3. It seems the fitting procedure used to determine the two fitting parameters is the least square fitting. Please provide the number and range of samples and the uncertainty levels of the fitting.”

Our answer

The number of samples (i.e. the number of measurement points) is given in the Methods (61 points). The range (30 mm in distance, 0.3 kHz in frequency) is displayed in Figure 2 (d and h). We have added in the Supplementary Information (Section S6) the uncertainty level of the fitting: "The standard error on these values can be roughly estimated from the covariance matrices obtained using `scipy.optimize.curve_fit`; the standard errors for the estimated effective diameters are 0.001 mm (water-steel interface) and 0.0005 mm (water-air interface); the standard errors for the estimated bubble-interface distances are 0.04 mm (water-steel interface) and 0.01 mm (water-air interface)."